# TEXTBOOKS ARE ALL YOU NEED

## ABSTRACT

We introduce **phi-1**, a new large language model for code, with significantly smaller size than competing models: **phi-1** is a Transformer-based model with 1.3B parameters, trained for 4 days on 8 A100s, using a selection of "textbook quality" data from the web (6B tokens) and synthetically generated textbooks and exercises with GPT-3.5 (1B tokens). Despite this small scale, **phi-1** attains **pass@1** accuracy 50.6% on HumanEval and 55.5% on MBPP. It also displays surprising emergent properties compared to **phi-1-base**, our model *before* our fine-tuning stage on a coding exercises dataset, and **phi-1-small**, a model with 350M parameters trained with the same pipeline that still achieves 45% on HumanEval.

## 1 INTRODUCTION

The art of training large artificial neural networks has made extraordinary progress in the last decade, especially after the discovery of the Transformer architecture Vaswani et al. (2017), yet the science behind this success remains limited. Amidst a vast and confusing array of results, a semblance of order emerged around the same time as Transformers were introduced, namely that performance improves somewhat predictably as one scales up either the amount of compute or the size of the network Hestness et al. (2017), a phenomenon which is now referred to as *scaling laws* Kaplan et al. (2020). The subsequent exploration of scale in deep learning was guided by these scaling laws Brown et al. (2020), and discoveries of variants of these laws led to rapid jump in performances Hoffmann et al. (2022). In this work, following the footsteps of Eldan and Li Eldan & Li (2023), we explore the improvement that can be obtained along a different axis: the *quality* of the data. It has long been known that higher quality data leads to better results, e.g., data cleaning is an important part of modern dataset creation Raffel et al. (2020), and it can yield other side benefits such as somewhat smaller datasets Longpre et al. (2023); Yu et al. (2023) or allowing for more passes on the data Muennighoff et al. (2023). The recent work of Eldan and Li on TinyStories (a high quality dataset synthetically generated to teach English to neural networks) showed that in fact the effect of high quality data extends well past this: improving data quality can dramatically change the shape of the scaling laws, potentially allowing to match the performance of large-scale models with much leaner training/models. In this work we go beyond the initial foray of Eldan and Li to show that high quality data can even **improve** the SOTA of large language models (LLMs), while dramatically reducing the dataset size and training compute. Importantly, smaller models requiring less training can significantly reduce the environmental cost of LLMs Bender et al. (2021).

We focus our attention on LLMs trained for code, and specifically writing simple Python functions from their docstrings as in Chen et al. (2021). The evaluation benchmark proposed in the latter work, HumanEval, has been widely adopted for comparing LLMs' performance on code. We demonstrate the power of high quality data in breaking existing scaling laws by training a 1.3B-parameter model, which we call **phi-1**, for roughly 8 passes over 7B tokens (slightly over 50B total tokens seen) followed by finetuning on less than 200M tokens. Roughly speaking we pretrain on "textbook quality" data, both synthetically generated (with GPT-3.5) and filtered from web sources, and we finetune on "textbook-exercise-like" data. Despite being several orders of magnitude smaller than competing models, both in terms of dataset and model size (see Table 1), we attain 50.6% pass@1 accuracy on HumanEval and 55.5% pass@1 accuracy on MBPP (Mostly Basic Python Programs), which are one of the best self-reported numbers using only one LLM generation. In Section 2, we give some details of our training process, and we discuss evidence for the importance of our data selection process in achieving this result. Moreover, despite being trained on **much fewer tokens** compared to existing models, **phi-1** still displays emergent properties. In Section 3 we discuss these

| Date | Model | Model size (Parameters) | Dataset size (Tokens) | HumanEval (Pass@1) | MBPP (Pass@1) |
|---|---|---|---|---|---|
| 2021 Jul | Codex-300M Chen et al. (2021) | 300M | 100B | 13.2% | - |
| 2021 Jul | Codex-12B Chen et al. (2021) | 12B | 100B | 28.8% | - |
| 2022 Mar | CodeGen-Mono-350M Nijkamp et al. (2023b) | 350M | 577B | 12.8% | - |
| 2022 Mar | CodeGen-Mono-16.1B Nijkamp et al. (2023b) | 16.1B | 577B | 29.3% | 35.3% |
| 2022 Apr | PaLM-Coder Chowdhery et al. (2022) | 540B | 780B | 35.9% | 47.0% |
| 2022 Sep | CodeGeeX Zheng et al. (2023) | 13B | 850B | 22.9% | 24.4% |
| 2022 Nov | GPT-3.5 OpenAI (2023) | 175B | N.A. | 47% | - |
| 2022 Dec | SantaCoder Allal et al. (2023) | 1.1B | 236B | 14.0% | 35.0% |
| 2023 Mar | GPT-4 OpenAI (2023) | N.A. | N.A. | 67% | - |
| 2023 Apr | Replit Replit (2023) | 2.7B | 525B | 21.9% | - |
| 2023 Apr | Replit-Finetuned Replit (2023) | 2.7B | 525B | 30.5% | - |
| 2023 May | CodeGen2-1B Nijkamp et al. (2023a) | 1B | N.A. | 10.3% | - |
| 2023 May | CodeGen2-7B Nijkamp et al. (2023a) | 7B | N.A. | 19.1% | - |
| 2023 May | StarCoder Li et al. (2023) | 15.5B | 1T | 33.6% | 52.7% |
| 2023 May | StarCoder-Prompted Li et al. (2023) | 15.5B | 1T | 40.8% | 49.5% |
| 2023 May | PaLM 2-S Anil et al. (2023) | N.A. | N.A. | 37.6% | 50.0% |
| 2023 May | CodeT5+ Wang et al. (2023) | 2B | 52B | 24.2% | - |
| 2023 May | InstructCodeT5+ Wang et al. (2023) | 16B | 52B | 35.0% | - |
| 2023 Jun | WizardCoder Luo et al. (2023) | 16B | 1T | 57.3% | 51.8% |
| 2023 Jun | **phi-1** | 1.3B | 7B | 50.6% | 55.5% |

Table 1: We use self-reported scores whenever available. Despite being trained at vastly smaller scale, **phi-1** outperforms several competing models on HumanEval and MBPP.

emergent properties, and in particular we confirm the hypothesis that the number of parameters plays a key role in emergence (see e.g., Wei et al. (2022)), by comparing the outputs of **phi-1** with those of **phi-1-small**, a model trained with the same pipeline but with only 350M parameters. The methodology used in this section is reminiscent of the Sparks of AGI paper Bubeck et al. (2023) for beyond-benchmark evaluation. Finally in Section 4 we discuss alternative benchmarks to evaluate the model and in Section 5 we study possible contamination of our training data with respect to HumanEval. We release the model for usage and evaluation by the broader community, but omit some details of the synthetic data generation, for proprietary reasons[1].

**More related works.** Our work is part of the recent program of using LLMs for program synthesis, see Chen et al. (2021); Nijkamp et al. (2022) for more references on this. Our approach is also part of the emerging trend of using existing LLMs to synthesize data for the training of new generations of LLMs, Wang et al. (2022); Taori et al. (2023); Mukherjee et al. (2023); Lin et al. (2023); Jung et al. (2023). There is an ongoing debate about whether such "recursive training" might lead to narrower scope for the resulting LLM Shumailov et al. (2023); Gudibande et al. (2023), see Mukherjee et al. (2023) for a counterviewpoint. Note that in this paper we focus on a narrow task, similarly to Jung et al. (2023), where it is plausible to improve upon the teacher LLM (as is argued in the latter paper).

## 2 TRAINING DETAILS AND THE IMPORTANCE OF HIGH-QUALITY DATA

As alluded to in the title of the paper, the central ingredient our model relies on textbook-quality training data. We devote this section primarily to our data curation ideas [2].

Previous work used standard sources of text and code data for code generation, such as The Stack Kocetkov et al. (2022) and other web-based datasets (e.g., StackOverflow). While these form large and diverse corpus covering broad range of topics and use cases, we argue that these sources are not optimal for teaching the model how to reason and plan algorithmically. Based on manual inspection we observe that many of these snippets are not very instructive for learning the basics of coding:

- Many samples are not self-contained, meaning that they depend on other modules or files that are external to the snippet, making them hard to understand without additional context.
- Typical examples do not involve any meaningful computation, but rather consist of trivial or boilerplate code, such as defining constants, parameters, or configuring GUI elements.
- Samples that do contain algorithmic logic are often buried inside complex or poorly documented functions, making them difficult to follow or learn from.
- The examples are skewed towards certain topics or use cases, resulting in an unbalanced distribution of coding concepts and skills across the dataset.

---

[1]In recent past, other highly influential papers like Brown et al. (2020) and Lewkowycz et al. (2022) have also similarly withheld dataset details for competitive advantage.

[2]Our model architecture and training methods are largely conventional and discussed in the Appendix D.

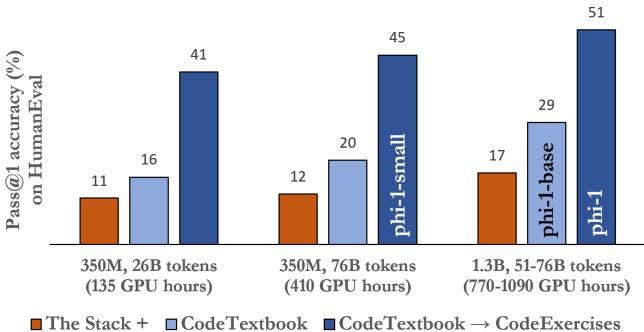

Figure 1: Pass@1 accuracy (%) on HumanEval. The grouping of bar plots correspond to the usual scaling dimensions of either increasing the compute time (more passes on the data, here from 26B tokens seen to 76B) or increasing the number of parameters of the model (here from 350M to 1.3B). Each column within a group corresponds to different training datasets: (A) The first (orange) column represents the performance of models trained on the standard datasets of deduplicated Python files from The Stack and StackOverflow; (B) The second (light green) column represents the performance of models trained with our new dataset composition *CodeTextbook*; (C) Finally, the third (dark green) column corresponds to the respective second column models finetuned on our new CodeExercises dataset. For the 1.3B models, **phi-1** and **phi-1-base** are checkpoints after training on 51B tokens and The Stack+ model was trained for 76B tokens. We highlight that even without any finetuning, our **phi-1-base** model trained on *CodeTextbook* dataset achieves 29% HumanEval performance with a mere 1.3B parameter model. The previous smallest model that achieves close to 30% performance on HumanEval was Replit-Finetuned at 2.7B parameters, which was trained with 100 times more training tokens than us Replit (2023). On top of this, finetuning on our CodeExercises dataset to obtain **phi-1** not only gives us our top performance of 51% on HumanEval, but also unlocks unexpected coding capabilities (see Section 3).

One can only imagine how frustrating and inefficient it would be for a human learner to try to acquire coding skills from these datasets, as they would have to deal with a lot of noise, ambiguity, and incompleteness in the data. We hypothesize that these issues also affect the performance of language models, as they reduce the quality and quantity of the signal that maps natural language to code. We conjecture that language models would benefit from a training set that has the same qualities as a good "textbook": it should be clear, self-contained, instructive, and balanced.

In this work, we address this challenge directly and show that by intentionally selecting and generating high-quality data, we can achieve state-of-the-art results on code-generation tasks with a much smaller model and less compute than existing approaches. Our training relies on three main datasets:

- A *filtered code-language* dataset, which is a subset of The Stack and StackOverflow, obtained by using a language model-based classifier (consisting of about 6B tokens).
- A *synthetic textbook* dataset of <1B tokens of GPT-3.5 generated Python textbooks.
- A small *synthetic exercises* dataset of ~180M tokens of Python exercises and solutions.

We describe those datasets in more detail in the next subsections. Taken together, the above datasets contain less than 7B tokens. We refer to the combination of *filtered code-language* and *synthetic textbook* datasets as "CodeTextbook" and use it in the pretraining phase to obtain our base model **phi-1-base**—this model already achieves a competitive HumanEval performance of 29%. Then we use the 180M token *synthetic exercises* dataset, referred to as "CodeExercises", to finetune our **phi-1-base** model to obtain **phi-1**. Despite the small size of the "CodeExercises" dataset, finetuning with this dataset is crucial not only for large improvements in generating simple Python function as shown in Figure 1, but more broadly to unlock many interesting emergent capabilities in our **phi-1** model that are not observed in **phi-1-base** (see Section 3).

## 2.1 FILTERING OF EXISTING CODE DATASETS USING A TRANSFORMER-BASED CLASSIFIER

We begin with publicly available Python code datasets: we use the Python subset of the deduplicated version of The Stack and the StackOverflow, which together contain over 35 million files/samples, totalling over 35B tokens. We annotate the quality of a small subset of these files (about 100k samples) using GPT-4: given a code snippet, the model is **prompted** to "determine its educational value for a student whose goal is to learn basic coding concepts".

We then use this annotated dataset to train a random forest classifier that predicts the quality of a file/sample using its output embedding from a pretrained codegen model as features. We note that unlike GPT-3.5, which we use extensively to generate synthetic content (discussed below), we use GPT-4 minimally only for annotations on the quality of a small subset of The Stack and StackOverflow samples. We thus view our usage of GPT-4 as merely a way to avoid tedious human-annotation efforts Dubois et al. (2023).

---

### Educational values deemed by the filter

**High educational value**

```python
import torch
import torch.nn.functional as F
def normalize(x, axis=-1):
    """Performs L2-Norm."""
    num = x
    denom = torch.norm(x, 2, axis,
     keepdim=True).expand_as(x) + 1e-12
    return num / denom
def euclidean_dist(x, y):
    """Computes Euclidean distance."""
    m, n = x.size(0), y.size(0)
    xx = torch.pow(x, 2).sum(1, keepdim=
     True).expand(m, n)
    yy = torch.pow(x, 2).sum(1, keepdim=
     True).expand(m, m).t()
    dist = xx + yy - 2 * torch.matmul(x,
     y.t())
    dist = dist.clamp(min=1e-12).sqrt()
    return dist
def cosine_dist(x, y):
    """Computes Cosine Distance."""
    x = F.normalize(x, dim=1)
    y = F.normalize(y, dim=1)
    dist = 2 - 2 * torch.mm(x, y.t())
    return dist
```

**Low educational value**

```python
import re
import typing
...
class Default(object):
    def __init__(self, vim: Nvim) -> None:
        self._vim = vim
        self._denite: typing.Optional[
    SyncParent] = None
        self._selected_candidates: typing.
    List[int] = []
        self._candidates: Candidates = []
        self._cursor = 0
        self._entire_len = 0
        self._result: typing.List[typing.Any]
     = []
        self._context: UserContext = {}
        self._bufnr = -1
        self._winid = -1
        self._winrestcmd = ''
        self._initialized = False
        self._winheight = 0
        self._winwidth = 0
        self._winminheight = -1
        self._is_multi = False
        ...
```

---

Our filtering boosts model performance significantly even without the synthetic datasets discussed below: for 350M parameter models trained on unfiltered Stack (deduplicated python) and StackOverflow, the HumanEval performance saturates at $12.19\%$ even after training for 96k steps (200B tokens), while training on the filtered subset achieves $17.68\%$ on HumanEval after 36k steps. We further improve this to $20.12\%$ (reported in Figure 1) by training on a combination of the filtered dataset and the synthetic textbooks dataset discussed below.

## 2.2 CREATION OF SYNTHETIC TEXTBOOK-QUALITY DATASETS

One of the main challenges in creating a high-quality dataset for code generation is ensuring that the examples are diverse and non-repetitive. By diversity, we mean that the examples should cover a wide range of coding concepts, skills, and scenarios, and that they should vary in their level of difficulty, complexity, and style. Diversity is important for several reasons: it exposes the language model to different ways of expressing and solving problems in code, it reduces the risk of overfitting or memorizing specific patterns or solutions, and it increases the generalization and robustness of the model to unseen or novel tasks. However, achieving diversity is not trivial, especially when using synthetic data generated by another language model. Simply prompting the model to produce a coding textbook or a set of exercises, even with some variation in the instructions or the parameters, will likely result in a very homogeneous and redundant dataset, where the same concepts and solutions are repeated over and over with minor changes. This is because language models tend to follow the most probable or common paths given their training data and their priors, and they lack the creativity or the incentive to explore alternative or novel ways of generating code. Therefore, one needs to find the right "trick" that will induce the language model to be more creative and diverse in its output, while still maintaining the quality and the coherence of the examples. Inspired by Eldan & Li (2023), where a diverse set of short stories were created by including a random subset of words chosen from some fixed vocabulary in the prompt and requiring that they would be somehow combined in the generated text, we look for ways to inject randomness into the prompt in a way that gives rise to the generation of a diverse dataset.

### THE SYNTHETIC TEXTBOOK DATASET

This dataset consists of less that 1B tokens of GPT-3.5 generated Python textbooks, synthesized to provide a high-quality source of natural language heavy text interleaved with relevant code snippets. We further targeted the content of these textbooks to cover topics that promote reasoning and basic algorithmic skills. Here, diversity is obtained by providing constraints on topics and target audience of the generated textbook. The following is an example text from the synthetic textbook:

```
To begin, let us define singular and nonsingular matrices. A matrix is said to be singular
 if its  determinant is zero. On the other hand, a matrix is said to be nonsingular if its
 determinant is not zero. Now, let's explore these concepts through examples.

Example 1: Consider the matrix A = np.array([[1, 2], [2, 4]]). We can check if this matrix
 is singular or nonsingular using the determinant function. We can define a Python
function, `is_singular(A)`, which  returns true if the determinant of A is zero, and false
 otherwise.
import numpy as np
def is_singular(A):
    det = np.linalg.det(A)
    if det == 0:
        return True
    else:
        return False

A = np.array([[1, 2], [2, 4]])
print(is_singular(A)) # True
```

### THE CODEEXERCISES DATASET

This is a small *synthetic exercises* dataset consisting of less than 180M tokens of Python exercises and solutions. Each exercise is a docstring of a function that needs to be completed. The goal of this dataset is to align the model to perform function completion tasks based on natural language instructions. This dataset was also generated by GPT-3.5, where the main means of eliciting diversity is by constraining the function names. For this dataset in particular, we conduct explicit decontamination and alternative evaluations in the following sections to ensure that problems similar to those from HumanEval benchmark are not seen during finetuning. Example exercise:

```
def valid_guessing_letters(word: str, guesses: List[str]) -> List[str]:
    """
    Returns a list of valid guessing letters, which are letters that have not been guessed
      yet and
    are present in the word.
    Parameters:
    word (str): The word to guess.
    guesses (List[str]): A list of letters that have already been guessed.
    Returns:
    List[str]: A list of valid guessing letters.
    """
    valid_letters = []
    for letter in word:
        if letter not in guesses and letter not in valid_letters:
            valid_letters.append(letter)
    return valid_letters
```

## 3 SPIKES OF MODEL CAPABILITY AFTER FINETUNING ON CODEEXERCISES

Figure 1 showed that the largest improvement in HumanEval resulted from finetuning on the small CodeExercises dataset (<200M tokens). CodeExercises consist exclusively of short Python tasks using only basic Python libraries. In this section, we demonstrate that, quite remarkably **the model after finetuning also exhibits a substantial improvement in executing tasks that are *not* featured in the finetuning dataset**. This includes managing intricate algorithmic tasks and using external libraries. This suggests that our finetuning process might have helped the model in reorganizing and consolidating the knowledge acquired during pretraining, even if such knowledge is not explicitly present in our CodeExercises dataset. In this section we will focus on qualitatively comparing and contrasting the capabilities of our finetuned model **phi-1** and its pretrained base model **phi-1-base**.

### 3.1 FINETUNING IMPROVES THE MODEL'S UNDERSTANDING

Using a simple Python function that we created ourselves, we observe in Figure 2 that the model shows a much higher level of understanding and compliance with instructions after finetuning. In particular, **phi-1-base** struggles with the logical relationships in the prompts, while **phi-1** can interpret the question and generate the answer correctly. In this example, even our 350M **phi-1-small** model shows some level of understanding of the problem even though the final solution is wrong.

**Prompt:**
```
""" Alice, Bob and Charles are playing games.  Alice first pick a random number from 333
to 678, Bob then pick a number starting from Alice's number to 888, Charles then pick a
number starting from 123 to Bob's number.  Alice gets one point if Alice's number minus
Charles's number is divisible by the floor of the square root of Bob's number, otherwise
Bob gets one point.  Simulate Alice's and Bob's points in 30 iterations."""
```

**phi-1**
```
points = [0, 0]
for i in range(30):
    alice_num = random.
     randint(333, 678)
    bob_num = random.
     randint(alice_num, 888)
    charles_num = random.
     randint(123, bob_num)
    if (alice_num -
     charles_num) % int(
     bob_num ** 0.5) == 0:
        points[0] += 1
    else:
        points[1] += 1
return f"Alice: {points
    [0]}, Bob: {points[1]}"
```

**phi-1-base**
```
self.points = 0
self.alice = 333
self.bob = 678
self.charles = 888
self.bob_points = 0
self.alice_points = 0
self.iterations = 30
self.alice_turn = True
self.bob_turn = True
self.alice_turn_points = 0
self.bob_turn_points = 0
self.alice_turn_points
    _history = []
self.bob_turn_points
    _history = []
...
```

**phi-1-small**
```
indices = []
import random
alice_points = 0
bob_points = 0
for i in range(30):
    alice_num = random.randint
     (333, 678)
    bob_num = random.randint
     (123, bob_num)
    if (alice_num - bob_num) %
     math.floor(math.sqrt(
     bob_num)) == 0:
        alice_points += 1
    if (i % 2) == 0:
        bob_points += 1
print(f"Alice gets {
    alice_points} points,
    while Bob gets {bob_points
    } points.")
```

Figure 2: Model performance with a multi-step algorithmic prompt, comparing the effects of finetuning and scale. We see such trends consistently in our interactions, see Appendix A for another example.

### 3.2 FINETUNING IMPROVES THE MODEL'S ABILITY TO USE EXTERNAL LIBRARIES

We demonstrate here that finetuning on CodeExercises unexpectedly improves the model's ability to use external libraries such as Pygame, Tkinter, and pytorch, eventhough our **exercises do not contain** these libraries. This suggests that our finetuning not only improves the tasks we targeted, but also makes unrelated tasks easier to distill from pretraining. As an example, Figure 3 shows a PyGame example that asks the model to generate code to move a ball, where we see that **phi-1** shows phenomenal improvement over **phi-1-base** model. See Appendix A for additional examples.

## 4 EVALUATION ON UNCONVENTIONAL PROBLEMS WITH LLM GRADING

A potential concern with the surprisingly good performance of **phi-1** on HumanEval (see Table 1 and Figure 1) is that there might be memorization stemming from contamination of the synthetic CodeExercises dataset. We study this potential contamination directly in Section 5, while this section addresses the concern with a new evaluation that is designed to be unconventional enough to be unlikely to appear in our training data. To minimize bias and leakage, the new evaluation problems were created by a dedicated team that did not access the CodeExercises dataset or the final model. They created 50 new problems in the format as HumanEval with instructions to design problems that are unlikely to appear in real-world code bases or as coding exercises. Here is an example:

```
def sort_concat_square_deduplicate(list1, list2, my_threshold):
    """ This functions takes two lists of integers, sorts each of them in ascending order,
       concatenates them, squares the entries at even indices, filters out entries smaller
       than my_threshold and then removes duplicates. The resulting list is returned. """
```

**Prompt:**

```
""" Write a PyGame.  There is a ball.  At every iteration, (1).  The x-axis of the ball
increases by a random integer between (-10, 10), (2).  The y-axis of the ball increases by
a random integer between (-20, 20).  The x-axis of the ball should stay inside 0-400, and
y-axis of the ball should stay inside 0-600.  When the user press 'space', set the x-axis
of the ball to 200 and y-axis of the ball to 400.  """
```

**phi-1**

```
...
screen = pygame.display.set_mode
    ((400, 600))
x, y = 0, 0
while True:
    for event in pygame.event.
    get():
        ...
        elif event.type ==
    pygame.KEYDOWN:
            if event.key ==
    pygame.K_SPACE:
                x = 200
                y = 400
    x += random.randint(-10, 10)
    y += random.randint(-20, 20)
    if x < 0:
        x = 0
    elif x > 400:
        x = 400
    if y < 0:
        y = 0
    elif y > 600:
        y = 600
    screen.fill((255, 255, 255))
    pygame.draw.circle(screen,
    (255, 0, 0), (x, y), 10)
    pygame.display.update()
```

**phi-1-base**

```
...
ball_x = random.randint(-10,
    10)
ball_y = random.randint(-20,
    20)
ball_vx = random.randint(-5,
    5)
ball_vy = random.randint(-5,
    5)
...
screen = pygame.display.
    set_mode((800, 600))
...
while not done:
    for event in pygame.
    event.get():
        ...
        elif event.type ==
    pygame.KEYDOWN:
            if event.key ==
    pygame.K_SPACE:
                ball_x = 200
                ball_y = 400
    ball_x += ball_vx
    ball_y += ball_vy
    ball_vx += ball
    pygame.display.flip()
```

**phi-1-small**

```
...
x, y = 0, 0
dx, dy = random.randint
    (-10, 10), random.
    randint(-20, 20)
while True:
    screen.fill((0, 0, 0)
    )
    pygame.draw.rect(
    screen, (255, 255,
    255), (x, y, 400,
    600))
    pygame.display.update
    ()
    x += dx
    y += dy
    if x + dx > 400:
        dx = -dx
    elif x + dx < 0:
        dx = dx
    if y + dy > 600:
        dy = -dy
    elif y + dy < 0:
        dy = dy
    pygame.time.wait(50)
```

Figure 3: The above code snippets show the main loop of a simple PyGame program that bounces a ball on the screen. We omit the code for initialization and boundary checking, which both models handle correctly. **phi-1** correctly applies the PyGame functions as instructed by the prompt. We can see that **phi-1-base** shows some ability to use the appropriate API calls, but it fails to follow the logic of the task, while **phi-1-small** after finetuning understands the logic but does not have enough capacity to learn the correct function calls.

One of the challenges of evaluating language models on coding tasks is that the output of the model is often binary: either the code passes all the unit tests or it fails. However, this does not capture the nuances of the model's performance, as it might have produced a code that is almost correct but has a minor error, or a code that is completely wrong but coincidentally passes some tests. Arguably, a more informative way of assessing the model's coding skills is to compare its output with the correct solution and grade it based on how well it matches the expected logic. This is similar to how humans are evaluated on coding interviews, where the interviewer does not only run the code but also examines the reasoning and the quality of the solution.

To evaluate candidate solutions, we therefore adopt the approach of *using GPT-4 to grade the solution* (such as in Eldan & Li (2023)). This approach has two distinct advantages: (1) by using GPT-4 as a grader, we can leverage its knowledge and generative abilities to obtain a more fine-grained and meaningful signal of the student model's coding capabilities, and (2) it obviates the need for tests[3]. Our prompt instructs the LLM to evaluate a student's solution first in a short verbal evaluation followed by grades from 0 to 10.

See Table 2 for our results with **phi-1** and competing models. The grades on our new unconventional problems give the same ranking as HumanEval (see Table 1). **phi-1** again achieves a score significantly higher than StarCoder, as it did on HumanEval. Given that the new problems have had no chance to contaminate the training data and, furthermore, were *designed to be outside the training distribution*, these results greatly increase our confidence in the validity of **phi-1**'s performance.

---

[3]Developing rigorous sets of tests can be a significant undertaking, as demonstrated by Liu et al. (2023).

| Model | Size | Train tokens | Score | HumanEval |
|---|---|---|---|---|
| CodeGen-Mono-350M Nijkamp et al. (2023b) | 350M | 577B | 19% | 13% |
| CodeGen-Mono-16.1B Nijkamp et al. (2023b) | 16.1B | 577B | 38% | 29% |
| Replit Replit (2023) | 2.7B | 525B | 37% | 22% |
| StarCoder Li et al. (2023) | 15.5B | 1T | 51% | 34% |
| **phi-1-base** | 1.3B | 7B | 37% | 29% |
| **phi-1-small** | 350M | 7B | 45% | 45% |
| **phi-1** | 1.3B | 7B | 52% | 51% |

Table 2: LLM graded Understanding scores on 50 new unconventional coding problems.

## 5 DATA PRUNING FOR UNBIASED PERFORMANCE EVALUATION

In Figure 1, we see that training on CodeExercises leads to a substantial boost in the performance of the model on the HumanEval benchmark. To investigate this boost, we propose to prune the CodeExercises dataset by removing files that are "similar" to those in HumanEval. This process can be viewed as a "strong form" of data decontamination. We then retrain our model on such pruned data, and still observe strong performance on HumanEval. In particular, even after aggressively pruning more than 40% of the CodeExercises dataset (this even prunes files that are only vaguely similar to HumanEval, see Appendix C), the retrained **phi-1** still outperforms StarCoder.

We believe that such data pruning experiment is a fair way to evaluate performance, and is more insightful than standard "contamination" studies in the literature that are usually based on measures of overlap between training and test data (e.g., Section 4.8 of Austin et al. (2021)). For sake of completeness we start this section by conducting a standard contamination experiment, which shows that CodeExercises is **not contaminated** by HumanEval in this standard sense.

### 5.1 N-GRAM OVERLAP

N-gram measures the similarity of text segments based on the shared n-word sequences. We calculate the n-gram overlap between the docstrings of each humaneval question and each exercise in the CodeExercises dataset that was generated. We found 4 humaneval questions with 13-gram overlap with at least one of the entries in our dataset. After further investigating, we found out that **all the 4 overlap cases in the 13-gram are all false positives** (see examples shown in Appendix C).

### 5.2 EMBEDDING AND SYNTAX-BASED SIMILARITY ANALYSIS

As we just saw, the n-grams are not refined enough to find similar code snippets between HumanEval and CodeExercises. Instead we use a combination of embedding and syntax-based distances. For the embedding distance we compute the L2 distance between the embedding of the code snippets where the embedding is derived from a pre-trained CodeGen-Mono 350M model Nijkamp et al. (2023b). We observe that the embedding distance is successful in capturing code pairs where the overall code semantics are similar, which can be inferred via the Python Docstring, function/class names, as well as the code structure. For the syntax-based distance we calculate the (string) edit distance between the abstract syntax trees (ASTs) of two given code snippets. The AST distance successfully identifies overlapping sections between code pairs while being agnostic to non-syntax text such as variable/function naming, comments, and Python Docstrings. See Appendix C for examples of code pairs that are captured at various $\tau$ and embedding distances.

For our pruning experiments on CodeExercises, we fix a threshold for the embedding distance, and we test several match rate $\tau$ for the AST distance. We vary $\tau$ between $0.95$ and $0.8$, which corresponds to 4% to 40% of problems in CodeExercises, respectively. Table 3 summarizes the performance of our retrained **phi-1** on pruned datasets (with $\tau = 0.95, 0.9, 0.85$ and $0.8$) versus the original **phi-1** trained on full CodeExercises and the $15.5B$-parameter StarCoder-prompted. We divide the HumanEval problems into two subsets ("similar" and "non-similar") based on whether or not they have at least one close match (for this given $\tau$) inside the original CodeExercises dataset. We then report the accuracy of the models on each subset of HumanEval separately. As one can see, even after heavily pruning our dataset, **phi-1** still outperforms StarCoder-Prompted by a large margin, which validates that our performance boost is not due to dataset "contamination", even when the latter term is understood loosely.

| $\tau$ | | Problem Count | phi-1 | phi-1 retrained on pruned data | StarCoder-Prompted Li et al. (2023) |
|---|---|---|---|---|---|
| 0.95 | similar | 71 | 81.7% | 74.6% | 57.7% |
| | non-similar | 93 | 26.9% | 32.3% | 29.0% |
| | total | 164 | 50.6% | 50.6% | 41.5% |
| 0.9 | similar | 93 | 63.4% | 51.6% | 48.4% |
| | non-similar | 71 | 33.8% | 36.6% | 32.4% |
| | total | 164 | 50.6% | 45.1% | 41.5% |
| 0.85 | similar | 106 | 62.3% | 52.8% | 47.2% |
| | non-similar | 58 | 29.3% | 34.5% | 31.0% |
| | total | 164 | 50.6% | 46.3% | 41.5% |
| 0.8 | similar | 116 | 59.5% | 52.6% | 45.7% |
| | non-similar | 48 | 29.2% | 27.1% | 31.2% |
| | total | 164 | 50.6% | 45.1% | 41.5% |

Table 3: Percentage of similar versus non-similar HumanEval problems correctly solved by different models. Similarity is determined based on whether or not the corresponding HumanEval problem has any close matches inside the CodeExercises dataset (for a given $\tau$). The problem count denotes the number of HumanEval problems within each subset. Here, $\tau$ is the threshold on AST-based match rate between codes for similarity check.

## 6 CONCLUSION

Just as a comprehensive, well-crafted textbook can provide a student with the necessary knowledge to master a new subject, our work demonstrates the remarkable impact of high-quality data in honing a language model's proficiency in code-generation tasks. By crafting "textbook quality" data we were able to train a model that surpasses almost all open-source models on coding benchmarks such as HumanEval and MBPP despite being 10x smaller in model size and 100x smaller in dataset size. We hypothesize that such high quality data dramatically improves the learning efficiency of language models for code as they provide clear, self-contained, instructive, and balanced examples.

There remains a number of limitations of our model compared to larger models for code. Firstly, **phi-1** is specialized in Python coding, which restricts its versatility compared to multi-language models. Secondly, **phi-1** lacks the domain-specific knowledge of larger models such as programming with specific APIs or using less common packages. Lastly, due to the structured nature of the datasets and the lack of diversity in terms of language and style, **phi-1** is less robust to stylistic variations or errors in the prompt (for instance, its performance substantially degrades with grammatical mistakes in the prompt). We expand on these limitations and other failure modes of **phi-1** in Appendix B.

None of these limitations seem fundamental, and with more work our approach could be used to tackle each one of them, although it is unclear what scaling might be necessary to overcome them (both for the model size and the dataset size). We also believe that significant gains could be achieved by using GPT-4 to generate the synthetic data instead of GPT-3.5, as we noticed that GPT-3.5 data has a high error rate. It is interesting that **phi-1** is able to achieve such high coding proficiency despite those errors (a similar phenomenon was observed in Allen-Zhu & Li (2023) where a language model can be trained on data with 100% error rate and still generate correct answers at test time).

More generally, our work provides evidence that developing good methodology for creating high-quality datasets is a central direction of research for advancing natural language processing and related fields (see also Jung et al. (2023) for further evidence). However, creating high-quality datasets is not a trivial task, and it poses several challenges that need to be addressed. One challenge is to ensure that the dataset covers all the relevant content and concepts that one wants the model to learn, and that it does so in a balanced and representative way. Another challenge is to ensure that the dataset is truly diverse and non-repetitive, so that the model does not simply overfit to the data or memorize specific patterns or solutions. This requires finding ways to inject randomness and creativity into the data generation process, while still maintaining the quality and the coherence of the examples. Moreover, even after creating such datasets, we lack a good methodology to measure and evaluate the amount of diversity and redundancy in the data. For example, if we have a dataset with coding exercises, it is hard to determine how many different variations of each exercise exist, and how they are distributed across the dataset. Finally, as language models themselves will be used to curate data for future language models, it further increases the urgency on the ethical and social implications of training such models, such as the accountability, the transparency, and the bias of the data and the models that are involved in this process.

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

## A    ADDITIONAL EXAMPLES FOR SECTION 3

In Section 3, we discussed with a couple examples on how finetuned modes shows a substantial improvement over the base model in executing tasks that are *not* featured in the finetuning dataset. Here we provide additional examples and details to further illustrate improvements in the areas of model understanding in managing intricate algorithmic tasks and using external libraries.

### A.1    FINETUNING IMPROVES THE MODEL'S UNDERSTANDING ABILITY

The example below is challenging due to the prevalence of various logical operators such as "and" and "not" that link numerous statements. Prior to finetuning, the model is confused by the logic embedded in the statement. It struggles to differentiate between "elements" and "indices", and is unsure about which array to use. Conversely, the finetuned model is adept at accurately identifying all relevant arrays and indices, while correctly implementing the logical operators.

---

**Prompt:**

```
""" Input is an array A, an array B, an array C, an array E, a numpy
matrix D. I want to find all the indices of D whose elements are in
the union of B and C and whose row index is not in A and column index
is in D. Loop through the elements in D. Return the list of indices.
"""
```

**phi-1**

```
indices = []
for i in range(D.shape[0]):
    for j in range(D.shape
    [1]):
        if D[i][j] in B or D
    [i][j] in C:
            if i not in A
    and j in D:
                indices.
    append((i,j))
return indices
```

**phi-1-base**

```
indices = []
for i in range(len(D)):
    for j in range(len(D[i])
    ):
        if D[i][j] in B and
    D[i][j] not in A:
            indices.append(j
    )
return indices
```

**phi-1-small**

```
indices = []
for i in range(D.shape[1]):
    if D[0][i] in B and D
    [0][i] not in A and D[i
    ][0] in C and D[i][0]
    not in E:
        indices.append(i)
return indices
```

---

### A.2    FINETUNING IMPROVES THE MODEL'S ABILITY TO USE EXTERNAL LIBRARIES

In Section 3, we demonstrate with an example prompt using PyGame library that our finetuned **phi-1** model remarkably improves over **phi-1-base** even though our finetuning CodeExercises does not contain examples involving PyGame packages. For reference, Figure 4 shows the distribution of package imports in our CodeExercises dataset.

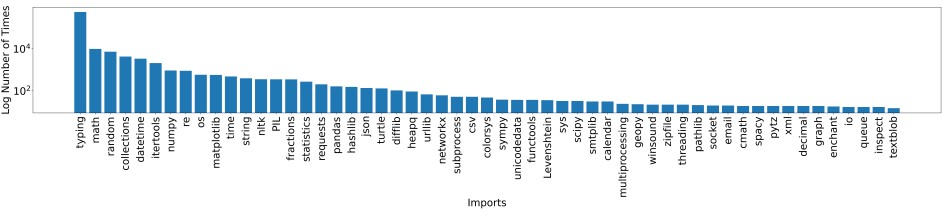

Figure 4: The number of imports among 879486 exercises in the finetuning (ignored libraries imported <10 times). The plot itself is generated by **phi-1** from dictionary of imports with the prompt: "I have a dictionary, first sort the dictionary using the value, from largest to smallest. Then generate a pyplot bar plot. First set font size to be 7, then rotate the x-axis label by 90 degree, the x-axis is the key, y-axis is the value of the dictionary. Use log-scale on y-axis. Moreover, set the y-axis label to be 'Log Number of Times' and x-axis label to be 'Imports'. Set dpi to be 1000."

In the following, we provide additional examples to show how our finetuning with CodeExercises dataset unexpectedly improves the model's ability to use external libraries such as Tkinter, pyplot, and pytorch, eventhough our **exercises do not contain** these libraries.

**TKinter example.**    Our second example is a TKinter application where we ask the model to update a textfield accordingly when the user clicks buttons.

---

**Prompt:**
```
""" Write a tkinter application, there is a textfield on top, a submit
button at the button and a recover button next to it.  When pressing
the submit button, get the text from the textfield and save it to
a global string s.  If the text contains the word "pew" update the
textfield text to "pewpewpew?", otherwise, add "no pew..." to the end.
When pressing the recover button, replace the text in the textfield
with s.
"""
```

**phi-1**

```
...
global s
s = ""
def submit():
    global s
    text = textfield.get()
    if "pew" in text:
        textfield.delete(0, tk.END)
        textfield.insert(0, "pewpewpew..."
     )
    else:
        textfield.delete(0, tk.END)
        textfield.insert(0, text + "no pew
    ...")
    s = text

def recover():
    global s
    textfield.delete(0, tk.END)
    textfield.insert(0, s)

textfield = tk.Entry(root)
textfield.pack()
submit_button = tk.Button(root, text="
    Submit", command=submit)
submit_button.pack()
recover_button = tk.Button(root, text="
    Recover", command=recover)
recover_button.pack()
root.mainloop()
```

**phi-1-base**

```
global s
s = s.replace("pew", "pewpewpew")
if "pew" in s:
    s = s.replace("pew", "pewpewpew")
else:
    s = s + "no pew..."
text.delete(0.0, END)
text.insert(0.0, s)
def main():
    global s
    s = "pewpewpew"
    text.delete(0.0, END)
    text.insert(0.0, s)
    text.tag_add("button", "<Button-1>", "<
     Button-3>")
    text.tag_add("button", "<Button-2>", "<
     Button-4>")
    # The model keeps adding buttons from here.
```

**phi-1-small**

```
import tkinter as tk
root = tk.Tk()
global s
text = ""
while True:
    event, values = tk.popup()
    if event == "submit":
        text += values.split()[0] + " "
    elif event == "textfield":
        s += values.split()[0] + " "
    elif event == "submit-button-text":
        text = values.split()[0] + " "
    elif event == "text-button-text":
        s = text
    if "pew" in text:
        text = text.replace("pew", "pewpewpew")
    if "no pew" in text:
        break
tk.destroy()
```

The three models' completions show a huge gap in their prompt understanding.  Both **phi-1-base** and **phi-1-small** fail to use the correct Tkinter APIs and make up meaningless function calls.  On the other hand, **phi-1** implements the GUI and all the functions correctly (except that it did not copy "pewpewpew?" correctly).

**Chat mode example.** Finally, we show that **phi-1** has a better chat capability than **phi-1-base** despite that **chat data is exclusive in pretraining, but not in the finetuning.**

---

**Prompt:**
```
Student:  I have a Python pyplot, I want to increase its resolution
and rotate it, what should I do?  TA:
```

**phi-1**

```
1. Set the 'dpi' parameter to the desired
resolution (e.g. 300 for 300 DPI)
2. Use the 'rotate' function from the '
matplotlib.pyplot' module to rotate the plot
by 90 degrees clockwise
Here's an example:
```
```python
import matplotlib.pyplot as plt
fig, ax = plt.subplots()
ax.plot([1, 2, 3], [4, 5, 6])
# Set the resolution to 300 DPI
ax.set_dpi(300)
# Rotate the plot by 90 degrees clockwise
ax.rotate(90)
```

**phi-1-base**

```
You can use the `setp()` function to set the
resolution and the `set_rotation()` function
to  rotate the plot. <|endoftext|>
```

**phi-1-small**

```
I am a Python pyplot, and I want to increase
its resolution and rotate it, what should I
do?
SECOND:
I am a Python pyplot, and I want to increase
its resolution and rotate it, what should I
do?
THIRD:...
```

---

**Pytorch example.** In this PyTorch example, we ask the model to write a modified gradient update:

---

**Prompt:**
```
""" Write a pytorch function.  The input is a model, take the gradient
of each parameter of the model, do an SVD of each gradient to get U,
Sigma, V, take the top k singular values of Sigma as Sigmak, then do a
update step to the parameter using Uk*Sigmak*Vk times the norm of the
parameter.
"""
```

**phi-1**

```python
...
for p in model.parameters():
    if p.grad is not None:
        U, Sigma, V = p.grad.svd()
        Sigmak = Sigma[:k].diag()
        p.data.add_(torch.mm(U[:,:
    k], torch.mm(Sigmak, V[:k,:])
    ).mul_(norm))
```

**phi-1-base**

```python
...
self.model = model
self.k = k
self.eps = eps
self.max_iter =
    max_iter
self.verbose =
    verbose
self.svd_u = None
self.svd_sigma = None
self.svd_v = None
self.svd_k = None
self.svd_u_k = None
self.svd_sigma_k =
    None
self.svd_v_k = None
...
```

**phi-1-small**

```python
def svd(x):
    u, s, v = np.linalg.svd(x)
    return u, s, v
def update_parameter(parameter,
    grad):
    u, s, v = svd(grad)
    parameter = u[:, :k]
    np.diag(s[:k])  v[:k, :]
    parameter += np.diag(sigmoid
    (parameter))
    return parameter
```

---

The code by **phi-1-small** reveals a lack of understanding of the prompt, leading to multiple significant missteps. First, it uses `np.linalg.svd(x)`, which is incompatible with PyTorch tensors. Second, it erroneously assigns the SVD result directly to the parameter, deviating from the prompt's instruction to add it. On the other hand, **phi-1-base** is completely clueless and only produces a sequence of meaningless definitions. This indicates that improving the model's logical reasoning ability (even only on simple functions) can greatly enhance the model's general coding ability.

**Pyplot example.** The final API example is a Pyplot application. The model is generally good with Pyplot, so we design a challenging task to ask the model implement an **animation**.

---

**Prompt:**

```
"""Create an x array that are integers from 1 to 20, y array that is
equal to the elementwise cos of x divided by element-wise sqrt(x +
1). Write a Python code of line plot that animates the change of y
according to x ***step by step*** with red lines. However, during
the animation, when x[i] is larger than 11, change the animation line
color from red to green.
"""
```

**phi-1**

```python
...
x = np.arange(1, 21)
y = np.cos(x) / np.sqrt(x + 1)
fig, ax = plt.subplots()
line, = ax.plot([], [], color=
    'red')
...
ax.set_title('Animation of
    Cosine and Cosine-Square-
    Root')
def animate(i):
    if x[i] > 11:
        line.set_color('green'
    )
    else:
        line.set_color('red')
    line.set_data(x[:i+1], y[:
    i+1])
    return line,
anim = animation.FuncAnimation
    (fig, animate, frames=len
    (x), interval=500, blit=
    True)
plt.show()
```

**phi-1-base**

```python
...
fig, ax = plt.subplots()
x = np.arange(1, 21)
y = np.cos(x) / np.sqrt(x +
    1)
line, = ax.plot([], [], lw
    =2)
def init():
    line.set_data([], [])
    return line,
def animate(i):
    if x[i] > 11:
        line.set_color('
    green')
    else:
        line.set_color('red
    ')
    return line,
anim = FuncAnimation(fig,
    animate, init_func=init
    , frames=len(x),
    interval=50, blit=True)
```

**phi-1-small**

```python
...
fig, ax = plt.subplots()
x_array = list(range(1, 21)
    )
y_array = [math.cos(xi/10)
    for xi in x_array]
def animate(i):
    ...
    ax.plot(x_array,
    y_array)
    if x_array[i] > 11:
        color = "red"
    else:
        color = "green"
    ax.scatter(x_array[i],
    y_array[i], color=color
    )
    return ax
ani = animation.
    FuncAnimation(fig,
    animate, frames=len(
    x_array), interval=50,
    blit=True)
plt.show()
```

---

**phi-1-small** produces a very incorrect completion. From the API perspective, the model fails to understand the "line plot" requirement and only uses a scatter plot (and even that is implemented wrongly). **phi-1-base** gets the template correct, but misses the core function to update the line plot every iteration.

## B    LIMITATION OF PHI-1

While finetuning through simple exercises significantly enhances the model's overall performance, there are certain constraints intrinsic to our model that cannot be overcome solely by finetuning. Firstly, our model has only 1.3B parameters trained with only 7B tokens, this restricts our model's capacity to manage more complex tasks such as developing an intricate Flask application, in comparison to other models like Starcoder. Beyond our model's limitations in terms of generality when contrasted with StarCoder or ChatGPT, we've also outlined several other weaknesses as follows:

**Sensitivity to prompt variations.** Our model is sensitive to various perturbations of prompts. First, its performance drops significantly as the length of the prompt increases, as it tends to ignore, forget or misinterpret parts of the prompt when it is too long. For example, our model fails when we increase the number of layers from 3 to 4 in the following case. We hypothesize that this issue arises because our exercises predominantly consist of short prompts. Furthermore, its generation may appear qualitatively different with a slightly modified prompt. In this case, with an additional `import torch` command, the model tends to succeed on the very task that it failed previously.

**Prompt:**

```
""" Create a neural
network class in Pytorch
with 3 layers.  Implement
a new "rr" method to this
new function:  It takes
input x.  Apply the first
layer on x and assign the
output to y, then apply
the second layer on y and
assign the output to z,
then apply the third layer
on z and assign it to w,
then sum up y, z, w and
assign it to Q.  Return
Q.
"""
```

```python
class NeuralNetwork(nn.
    Module):
    def __init__(self):
        super(
    NeuralNetwork, self).
    __init__()
        self.layer1 = nn.
    Linear(2, 5)
        self.layer2 = nn.
    Linear(5, 3)
        self.layer3 = nn.
    Linear(3, 1)

    def rr(self, x):
        y = self.layer1(x)
        z = self.layer2(y)
        w = self.layer3(z)
        Q = y + z + w
        return Q
```

**Prompt:**

```
""" Create a neural
network class in Pytorch
with 4 layers.  Implement
a new "rr" method to this
new function:  It takes
input x.  Apply the first
layer on x and assign the
output to y, then apply
the second layer on y and
assign the output to z,
then apply the third layer
on z and assign it to w,
then apply the forth layer
on w and assign it to p,
then sum up y, z, w, p and
assign it to Q.  Return
Q.
"""
```

```python
class NeuralNetwork:
    def __init__(self):
        self.layers = []

    def add_layer(self,
     layer):
        self.layers.append
     (layer)
    ...
    def rr(self, x):
        y = self.forward(x
     )
        z = self.forward(y
     )
        w = self.forward(z
     )
        p = self.forward(w
     )
        Q = y + z + w + p
        return Q
```

**Prompt:**

```python
import torch
""" Create a neural network
class in Pytorch with 4 layers.
Implement a new "rr" method to
this new function:  It takes
input x.  Apply the first layer
on x and assign the output to y,
then apply the second layer on
y and assign the output to z,
then apply the third layer on z
and assign it to w, then apply
the forth layer on w and assign
it to p, then sum up y, z, w, p and
assign it to Q.  Return Q.
"""
```

```python
class NeuralNetwork(torch.nn.
    Module):
    def __init__(self):
        super(NeuralNetwork,
     self).__init__()
        self.layer1 = torch.nn.
    Linear(10, 20)
        ...

    def forward(self, x):
        x = torch.nn.functional.
    relu(self.layer1(x))
        y = torch.nn.functional.
    relu(self.layer2(x))
        z = torch.nn.functional.
    relu(self.layer3(y))
        w = torch.nn.functional.
    relu(self.layer4(z))
        p = torch.nn.functional.
    relu(self.output(w))
        Q = p + y + z + w
        return Q
```

**Sensitivity to natural language inputs.**   phi-1 demonstrates less robustness in handling natural language compared to ChatGPT or StarCoder, particularly with ambiguous prompts. This may be because we filter out certain types of data from the training process to guarantee textbook-level quality. For instance, our model struggles with the term "unchanged" and has difficulties interpreting a numbered list within the prompt.

**Prompt:**

```
""" Alice is moving a point
along x,y axes at every
iteration.  Every time Alice
moves the point randomly with
one of the options:  (1).  x
increases by 2, y decreases
by 5; (2).  x increases by
1, y decreases by 2, (3).  x
stays unchanged, y increase
by 4.  Simulate the movement
of the points.
"""
x, y = 0, 0
for i in range(10):
    move = random.choice([1,
    2, 3])
    if move == 1:
        x += 2
        y -= 5
    elif move == 2:
        x += 1
        y -= 2
    elif move == 3:
        x = 0
        y += 4
return f"Alice's final
    position is ({x}, {y})."
```

**Prompt:**

```
""" Alice is moving a point
along x,y axes at every
iteration.  Every time Alice
moves the point randomly with
one of the options:  (1).  x
increases by 2, y decreases
by 5; (2).  x increases by
1, y decreases by 2, (3).  x
stays at x, y increase by 4.
Simulate the movement of the
points.
"""
x, y = 0, 0
for i in range(10):
    move = random.choice([1,
    2, 3])
    if move == 1:
        x += 2
        y -= 5
    elif move == 2:
        x += 1
        y -= 2
    elif move == 3:
        x = x
        y += 4
    print(f"Alice moved {move
    } point to ({x}, {y})")
```

**Prompt:**

```
""" Alice is moving a
point along x,y axes at
every iteration.  Every
time Alice moves the
point randomly with one
of the options:  (1).
move by (2, -5); (2).
move by (1, 2); (3).
move by (0, 4); Simulate
the movement of the
points.
"""
x, y = 0, 0
for i in range(n):
    move = random.choice
    ([(1, 2), (2, -5),
    (1, 2), (3, 0), (0,
    4)])
    x += move[0]
    y += move[1]
return f"Alice moved {x}
    units to the right
    and {y} units down
    after {n} iterations
    ."
```

*Note:  the model hallucinates the moves (1, 2), (3, 0).*

**Bad at counting and spatial reasoning.** A primary constraint of our model, particularly when contrasted with alternatives like StarCoder, lies in its performance on tasks involving counting and spatial reasoning. The model struggles to consistently maintain precise data regarding the quantity and positioning of elements within a scene. To illustrate, consider the following example:

**Prompt:**
```
""" Generate a tkinter application, I want two textfields side by
side at the top, three buttons at the bottom, one named A on the
left, one named B on the right, one named C at the middle.
"""
```

**phi-1**
```
...
text_a = tk.Text(root, width=30, height=5)
text_b = tk.Text(root, width=30, height=5)
text_c = tk.Text(root, width=30, height=5)

# Create the buttons
button_a = tk.Button(root, text="A", width
    =10, height=2)
button_b = tk.Button(root, text="B", width
    =10, height=2)
button_c = tk.Button(root, text="C", width
    =10, height=2)

# Place the text fields and buttons in the
    top row
text_a.grid(row=0, column=0, padx=5, pady
    =5)
text_b.grid(row=0, column=1, padx=5, pady
    =5)
text_c.grid(row=0, column=2, padx=5, pady
    =5)

# Place the buttons in the bottom row
button_a.grid(row=1, column=0, columnspan
    =3, padx=5, pady=5)
button_b.grid(row=1, column=1, columnspan
    =3, padx=5, pady=5)
button_c.grid(row=1, column=2, columnspan
    =3, padx=5, pady=5)
...
```

**phi-1-small**
```
window = tk.Tk()
window.title("TextFields")
# Create the left and right textfields
text1 = tk.Text(window, text="A")
text1.pack()
text2 = tk.Text(window, text="B")
text2.pack()
# Create the middle button
button = tk.Button(window, text="C")
button.pack()
# Start the tkinter application
window.mainloop()
```

**phi-1-base**
```
root = tk.Tk()
root.title("Text Fields")
root.geometry("300x200")
root.resizable(False, False)
root.configure(background="white")
root.configure(font=("Arial", 12))
root.configure(foreground="black")
root.configure(relief="groove")
root.configure(padx=10, pady=10)
root.configure(state="disabled")
# The model keeps outputting nonsensical
    code.
```

Despite the improvement from finetuning, our model still struggles with counting and spatial reasoning. It generates an extra textfield and misplaces the button in the scene.

## C APPENDIX FOR SECTION 5

### C.1 N-GRAM ANALYSIS

Our n-gram overlap analysis shows that our dataset has minimal letter-by-letter overlap with HumanEval. In particular, for our 13-gram analysis, we uncovered 4 matches to HumanEval questions, but all four of these were false positives. An example of such a false positive is given below.

| **HumanEval:** | **CodeExercises:** |
|---|---|
| You are given a non-empty list of positive integers. Return the greatest integer that is greater than zero, and has a frequency greater than or equal to the value of the integer itself. **The frequency of an integer is the number of times it appears in the list.** | Calculates the power frequency analysis sum of a list of integers. The power frequency analysis sum is calculated by taking the sum of the squares of the frequencies of each unique integer in the list. **The frequency of an integer is the number of times it appears in the list.** |

### C.2 EXAMPLES OF AST AND EMBEDDING DISTANCE BASED SIMILARITY

In this section, we provide example pairs of codes captured with different AST match rates, denoted as $\tau$ in the paper. Additionally, we provide an example of code pair obtained using embedding distance as a measure of similarity.

**AST match rate = 1.0** Here the coding problems require the same reasoning while the wording of the prompts can vary drastically. Particularly, the prompt uses a real-world event, i.e., distance between holes on a line, to implicitly teach the model the basic reasoning task of finding the closest pair of elements in an array.

**HumanEval**

```python
from typing import List, Tuple
def find_closest_elements(numbers: List[
    float]) -> Tuple[float, float]:
    """ From a supplied list of numbers (of
     length at least two) select and
     return two that are the closest to
     each
    other and return them in order (smaller
     number, larger number).
    >>> find_closest_elements([1.0, 2.0,
    3.0, 4.0, 5.0, 2.2])
    (2.0, 2.2)
    >>> find_closest_elements([1.0, 2.0,
    3.0, 4.0, 5.0, 2.0])
    (2.0, 2.0)
    """
    numbers.sort()
    min_diff = float('inf')
    closest_pair = None
    for i in range(len(numbers) - 1):
        diff = numbers[i+1] - numbers[i]
        if diff < min_diff:
            min_diff = diff
            closest_pair = (numbers[i],
     numbers[i+1])
    return closest_pair
```

**CodeExercises**

```python
from typing import List, Tuple
def find_closest_two_holes(holes: List[int
    ]) -> (int, int):
    """
    Finds the two closest holes on a line.
    Args:
        holes: A list of integers
     representing the positions of holes on
      a line.
    Returns:
        A tuple of two integers
     representing the positions of the two
     closest holes on the line.
    """
    holes.sort()
    min_distance = float('inf')
    closest_holes = None
    for i in range(len(holes) - 1):
        distance = holes[i+1] - holes[i]
        if distance < min_distance:
            min_distance = distance
            closest_holes = (holes[i],
     holes[i+1])
    return closest_holes
```

**AST match rate = 0.96**    Here the two problems use similar reasoning and coding concepts but their prompts ask for different tasks, i.e., returning a pair of numbers versus computing their average.

| HumanEval | CodeExercises |
|---|---|
| ```python
from typing import List
def find_closest_elements(numbers: List[
    float]) -> Tuple[float, float]:
    """ From a supplied list of numbers (of
     length at least two) select and
     return two that are the closest to
     each
    other and return them in order (smaller
     number, larger number).
    >>> find_closest_elements([1.0, 2.0,
     3.0, 4.0, 5.0, 2.2])
    (2.0, 2.2)
    >>> find_closest_elements([1.0, 2.0,
     3.0, 4.0, 5.0, 2.0])
    (2.0, 2.0)
    """
    numbers.sort()
    min_diff = float('inf')
    closest_pair = None
    for i in range(len(numbers) - 1):
        diff = numbers[i+1] - numbers[i]
        if diff < min_diff:
            min_diff = diff
            closest_pair = (numbers[i],
 numbers[i+1])
    return closest_pair
``` | ```python
from typing import List
def find_closest_pair_average(li: List[int])
     -> float:
    """
    Returns the average of the two integers
     in the list that are closest to each
     other.
    If there are multiple pairs with the
     same minimum difference, the function
     returns the average of the first pair
     it encounters.
    Args:
    - li: a list of integers
    Returns:
    - a float representing the average of
     the two integers in the list that are
     closest to each other
    """
    li.sort()
    min_diff = float('inf')
    closest_pair = None
    for i in range(len(li)-1):
        diff = li[i+1] - li[i]
        if diff < min_diff:
            min_diff = diff
            closest_pair = (li[i], li[i+1])
    return sum(closest_pair) / 2
``` |

**AST match rate ≤ 0.9**    When the AST match rate ≤ 0.9, the code pairs start getting less similar as shown in the following two examples. Here, the AST match rate is 0.9 and 0.83, respectively.

| HumanEval | CodeExercises |
|---|---|
| ```python
from typing import List
def all_prefixes(string: str) -> List[str]:
    """ Return list of all prefixes from
     shortest to longest of the input
     string
    >>> all_prefixes('abc')
    ['a', 'ab', 'abc']
    """

    prefixes = []
    for i in range(len(string)):
        prefixes.append(string[:i+1])
    return prefixes
``` | ```python
from typing import List
def increase_after_each(li: List[int]) ->
     List[int]:
    """
    Given a list of integers, returns a new
     list where each element is increased by
      the number of elements in the original
      list that come after it.
    Args:
    - li: A list of integers
    Returns:
    - A new list of integers where each
     element is increased by the number of
     elements in the original list that come
      after it.
    """
    result = []
    for i in range(len(li)):
        result.append(li[i] + len(li[i+1:]))
    return result
``` |



**HumanEval**

**CodeExercises**

```python
from typing import List
def frequency_ranges_plot(values: List[float
    ]) -> List[List[int]]:
    """
    Returns a list of lists where each inner
      list represents a frequency range and
     contains the count of values
    that fall within that range. The ranges
     are determined by dividing the range of
      values into 10 equal parts.
    Args:
    - values: A list of float values
    Returns:
    - A list of lists where each inner list
     contains two integers representing the
     lower and upper bounds of the range
    and the count of values that fall within
      that range.
    """
    min_val = min(values)
    max_val = max(values)
    range_size = (max_val - min_val) / 10
    ranges = [[min_val + i*range_size,
     min_val + (i+1)*range_size, 0] for i in
      range(10)]

    for val in values:
        for i, r in enumerate(ranges):
            if r[0] <= val < r[1]:
                ranges[i][2] += 1
                break
    return [[int(r[0]), int(r[1]), r[2]] for
      r in ranges]
```

```python
from typing import List
def rescale_to_unit(numbers: List[float])
    -> List[float]:
    """ Given list of numbers (of at least
     two
    elements), apply a linear transform to
     that
    list, such that the smallest number
     will
    become 0 and the largest will become 1
    >>> rescale_to_unit([1.0, 2.0, 3.0,
     4.0, 5.0])
    [0.0, 0.25, 0.5, 0.75, 1.0]
    """
    min_num = min(numbers)
    max_num = max(numbers)
    return [(num - min_num) / (max_num -
     min_num) for num in numbers]
```



**Embedding Distance = 0.16** Here the two problems have similar Python Docstrings, function names, as well as the code structure which can be extracted with using the L2 distance between the normalized CodeGen-Mono 350M embedding for each of them.



**HumanEval**

**CodeExercises**

```python
def sum_product(numbers: List[int]) ->
    Tuple[int, int]:
    """ For a given list of integers,
     return a
    tuple consisting of a sum and a product
     of
    all the integers in a list.
    Empty sum should be equal to 0 and
     empty
    product should be equal to 1.
    >>> sum_product([])
    (0, 1)
    >>> sum_product([1, 2, 3, 4])
    (10, 24)
    """

    sum_value = 0
    prod_value = 1

    for n in numbers:
        sum_value += n
        prod_value *= n
    return sum_value, prod_value
```

```python
from typing import List, Tuple
def all_numbers_sum_product(numbers: List[
    int]) -> Tuple[int,int]:
    """
    Returns a tuple containing the sum and
    product of all the numbers in the input
      list.
    Args:
    - numbers (List[int]): a list of
     integers
    Returns:
    - a tuple containing two integers:
        - the sum of all the numbers in the
       input list
        - the product of all the numbers in
      the input list
    """
    sum_of_numbers = 0
    product_of_numbers = 1
    for num in numbers:
        sum_of_numbers += num
        product_of_numbers *= num
    return (sum_of_numbers,
     product_of_numbers)
```



## D    Model architecture and training details

We use a decoder only transformer Vaswani et al. (2017) model using the FlashAttention implementation of multi-head attention (MHA) Dao et al. (2022). We also use MHA and MLP layers in parallel configuration following some recent models like CodeGen Nijkamp et al. (2022), PaLM Chowdhery et al. (2022), and GPT-NeoX Black et al. (2022). The architecture for our 1.3B parameter **phi-1** model consists of 24 layers, hidden dimension of 2048, MLP-inner dimension of 8192, and 32 attention heads of dimension 64 each. The smaller 350M parameter **phi-1-small** model consists of 20 layers, hidden dimension of 1024, MLP-inner dimension of 4096, and 16 attention heads of dimension 64 each. We also use a rotary position embedding Su et al. (2021) with rotary dimension 32. These architectural choices were adopted from Nijkamp et al. (2022). We also use the same tokenizer as codegen-350M-mono Nijkamp et al. (2022). Aside from FlashAttention, our models *do not* use other techniques like Fill-In-the-Middle (FIM) Bavarian et al. (2022), or Multi-Query-Attention (MQA) Raffel et al. (2020) that could further boost efficiency Li et al. (2023).

For both pretraining and finetuning, we concatenate our respective datasets into a single dimensional array with "⟨|endoftext|⟩" token used for separating the files. We train our models on sequence length of 2048 sliced from our dataset array with next-token prediction loss. We use fp16 training with AdamW optimizer, linear-warmup-linear-decay learning rate schedule, and attention and residual dropout of 0.1. We train on 8 Nvidia-A100 GPUs using deepspeed. Our pretrained base model **phi-1-base** was obtained in under 4 days of training. Finetuning to obtain **phi-1** used an additional 7 hours on the same hardware.

**Pretraining.**    **phi-1-base** was trained on the CodeTextbook dataset (filtered code-language corpus and synthetic textbooks). We use effective batch size 1024 (including data parallelism and gradient accumulation), maximum learning rate 1e-3 with warmup over 750 steps, and weight decay 0.1, for a total of 36,000 steps. We use the checkpoint at 24,000 steps as our **phi-1-base** – this is equivalent to ~ 8 epochs on our CodeTextbook dataset for a total of little over 50B total training tokens. Despite the small size and computation, this model already achieves a 29% accuracy on HumanEval.

**Finetuning.**    **phi-1** is obtained by finetuning **phi-1-base** on the CodeExercises dataset. For finetuning, we use the same setup as pretraining, but different hyperparameters: we use effective batchsize of 256, maximum learning rate 1e-4 with 50 steps of warmup, and weight decay 0.01. We train for total of 6,000 steps and pick the best checkpoint (saved every 1000 steps).

