# OpenReview forum: "Textbooks Are All You Need"
_ICLR.cc/2024/Conference — Submitted to ICLR 2024_

### Official Review · Reviewer_JG5J · 2023-10-26

**Soundness:** 3 good
**Presentation:** 4 excellent
**Contribution:** 3 good
**Rating:** 6
**Confidence:** 3

**Summary:**

This work explores the question of data quality, in terms of data with less noise and data well aligned with the tasks of interest, for training narrow Python code generation models . The authors introduce a set of small datasets totaling ~7B tokens consisting of (i) 6B token public code data filtered for quality using GPT4 and an embedding based classifier, (ii) <1B tokens of textbook-like data generated by GPT3.5, and (iii) 180M tokens of code completion exercises generated by GPT3.5. Pretraining and finetuning on these datasets yields highly performant models with limited size (1.3B parameters) evaluated on HumanEval and MBPP, with majority of the gain coming from finetuning on (iii).The authors also note some generalization capabilities beyond the limited datasets learned by their models.

**Strengths:**

The authors pursue a significant goal with this work, showing how large generalist models can be used to train high quality and efficient models with limited compute spend. The results are impressive on HumanEval - the 1.3B phi model (this work) outperforms much larger and compute intensive models.

**Weaknesses:**

The authors make a concerted effort to prove the "synthetic exercise" dataset curated in their work does not contain contaminated data from the test sets - however, I could not find a similar investigation for the "synthetic textbook" dataset.

Another significant weakness is that finetuning results for other models (e.g. StarCoder) are missing - these results can answer the question, "Are textbooks all you need or just the exercises?"

**Questions:**

It needs a study of other models, e.g. starcoder finetuned on their "synthetic exercises" dataset. How does their performance delta look?

Needs study of overlap of "synthetic textbook" data with test sets.

---

> ### Author Response · Authors · 2023-11-22
> **Author response**
>
> We thank the reviewer for their time and effort in reviewing our paper and highlighting our models impressive performance. Below is our response for addressing some of the concerns raised in the review
>
> **Q. The authors make a concerted effort to prove the "synthetic exercise" dataset curated in their work does not contain contaminated data from the test sets - however, I could not find a similar investigation for the "synthetic textbook" dataset.**
>
> We were able to do a thorough decontamination analysis for HumanEval on our CodeExercises dataset as both datasets were of the same type (short python functions) and are self-contained code snippets – this meant that we can devise robust versions of similarities beyond string matching. We note that many decontamination procedures in the literature (including The Stack dataset) resort to string-matching or min-hash matching against a test set which we show is not very robust (e.g., our analysis picked up similarities that N-gram analysis did not).
>
> On the other hand, our synthetic textbook is a fundamentally different NLP-heavy dataset. The code snippets in the text are often very short and used mostly for illustrative purposes: e.g, exercises or usage specific functions or structures like print and if-then, class definition etc. These are not directly comparable to HumanEval questions which have specific function level algorithmic style. Thus, we did not find this dataset suitable for in-depth ast-style analysis.
>
> **Q. Another significant weakness is that finetuning results for other models (e.g. StarCoder) are missing - these results can answer the question, "Are textbooks all you need or just the exercises?"**
>
> We emphasize that the benefit of “textbook” approach not only lies in the final performance but also in overall training compute. Compared to our model, all other baselines are pretrained with orders of magnitude larger compute. StarCode is 10x larger – and even finetuning on it is expensive computationally.  It is possible that CodeExercise finetuning will boost the performance of StarCoder and other models too, but that does not diminish the importance of “textbook” quality data for pretraining. We also highlight from our paper that even without any finetuning, our phi-1-base model trained on CodeTextbook dataset achieves 29% HumanEval performance with a mere 1.3B parameter model. The previous smallest model that achieves close to 30% performance on HumanEval was Replit-Finetuned at 2.7B parameters, which was trained with 100 times more training tokens than us Replit (2023).

---

> > ### Comment · Reviewer_JG5J · 2023-11-23
> >
> > I think the contamination study of the synthetic textbook dataset is required to accept the conclusions in this work, even if an in-depth analysis is difficult.
> >
> > > We also highlight from our paper that even without any finetuning, our phi-1-base model trained on CodeTextbook dataset achieves 29% HumanEval performance with a mere 1.3B parameter model. The previous smallest model that achieves close to 30% performance on HumanEval was Replit-Finetuned at 2.7B parameters, which was trained with 100 times more training tokens than us Replit (2023).
> >
> > This also makes me curious about any overlap between HumanEval and "synthetic textbooks", would love to see that ruled out.

---

### Official Review · Reviewer_DYsB · 2023-11-01

**Soundness:** 2 fair
**Presentation:** 4 excellent
**Contribution:** 2 fair
**Rating:** 6
**Confidence:** 3

**Summary:**

This paper introduces a series of new models named phi-1. Phi-1 is pre-trained on high-quality filtered code-related data, and further fine-tuned on GPT-3.5 generated code exercises. Phi-1 achieves impressive results on HumanEval and MBPP, surpassing competitive open-source code models such as StarCoder. The studies highlights the importance of high-quality training data.

**Strengths:**

* Phi-1 is a very strong code model given its size. The release of such model will be of interest to the community.
* The quality of training data is an important factor, and is previously overlooked. This paper highlights its importance and call for attention to it.
* Paper is written clearly and well organized.

**Weaknesses:**

I don't see strong weaknesses in this paper.
See clarification / discussion questions below.

**Questions:**

* Can you please briefly introduce the training data and method used by CodeGen-Mono, Replit and StarCoder? Will be helpful in understanding how Phi-1 is different and perhaps will further highlight the important of data quality.
* As mentioned in Sec 3.2, fine-tuning on coding exercises makes unrelated tasks easier to distill from pretraining. If I'm understanding it correctly, during pretraining, languages other than Python are used. Is it possible to further support this claim by evaluating on non-Python coding tasks?
* It was mentioned that Phi-1 has emergent abilities that were not observed in Phi-1-base. In Sec 3, examples were used to support this claim. Is it possible to evaluate this more systematically and qualitatively?
* Would be helpful to include GPT-4 performance on the 50 new unconventional coding problems.

---

> ### Author Response · Authors · 2023-11-22
> **Author response**
>
> We thank the reviewer for their time in reviewing our work and highlighting the impressiveness of our model. Please find the answers to your questions below.
>
> **Q. Can you please briefly introduce the training data and method used by CodeGen-Mono, Replit and StarCoder? Will be helpful in understanding how Phi-1 is different and perhaps will further highlight the important of data quality.**
>
> Thank you for raising the point. For Codegen-Mono we don’t know the exact dataset that was used in the training, but their paper states that they use combination of Python source code data from Github and NLP data from The Pile. Both these datasets are large corpuses of web data without additional filtering. StartCoder and Replit were trained on the deduplicated subset of The Stack (which in turn has github source code cleaned of PII and permissive licenses among other things). Both these models also use multiple code languages in pretraining. These models were trained on a total of 1T and 500B tokens, respectively.
>
> In comparison Phi-1, (a) we start with only the python subset of The Stack and stackoverflow, (b) but uses filtering to reduce the size to 20-25% of the original datasets (6B unique tokens), and (c) strategically add targetted synthetic data in pretraining and finetuning. We train on 7B unique tokens for slightly less than 8 epochs. Of the choices we made, the first choice of using python-only was not strategical and made for restricting the scope so we can do more controlled experiments. The other two choices were the new contributions in this work.
>
> **Q. As mentioned in Sec 3.2, fine-tuning on coding exercises makes unrelated tasks easier to distill from pretraining. If I'm understanding it correctly, during pretraining, languages other than Python are used. Is it possible to further support this claim by evaluating on non-Python coding tasks?**
>
> Unfortunately, our pretraining data had only python code snippets, but this would be a good experiment to expand on our model for multilingual code. This could certainly help distil information from other languages, although one consideration experimentally is that if finetuning is fully in python the model might be overfit to the python syntax. To avoid this, the right approach might be to mix in python CodeExercises with small amounts of filtered dataset from other languages – this way we can see the benefits of training with python-CodeExercises on other languages without being confounded by the syntax overfitting.
>
> **Q. It was mentioned that Phi-1 has emergent abilities that were not observed in Phi-1-base. In Sec 3, examples were used to support this claim. Is it possible to evaluate this more systematically and qualitatively?**
>
> The main surprising emergent ability we found was that the model got better at API usage of popular apis even if the finetuning dataset did not have those API’s. Unfortunately, there are no standard benchmarks we could find for evaluating API usage, so we show more qualitative examples of the correct usage.
>
> **Q. Would be helpful to include GPT-4 performance on the 50 new unconventional coding problems.**
>
> Thank you for suggesting it. We will add this number. Below is the updated Table.
>
> | Model | GPT-Score | HumanEval |
> |---|---|---|
> | CodeGen-Mono-16.1B Nijkamp et al. (2023b) | 38% | 29% |
> | Replit Replit (2023) | 37% | 22% |
> | StarCoder Li et al. (2023) | 51% | 34% |
> | phi-1-base | 37% | 29% |
> | phi-1 | 52% | 51% |
> |GPT-4 | 73%| 67% (reported)|

---

### Official Review · Reviewer_UYV3 · 2023-11-01

**Soundness:** 2 fair
**Presentation:** 3 good
**Contribution:** 3 good
**Rating:** 6
**Confidence:** 4

**Summary:**

The paper introduces phi-1, a new large language model for code, that achieves impressive performance on coding tasks despite its smaller size compared to competing models. The authors highlight the importance of high-quality data in improving the performance of language models for code generation tasks. They provide evidence of the benefits of using textbook-quality data and demonstrate the effectiveness of their approach through empirical evaluations.

**Strengths:**

1) Innovative methodology: The paper proposes the use of high-quality data, specifically textbook-quality data, to train a language model for code generation. This approach is novel and demonstrates the impact of data quality on model performance.

2) Insightful empirical findings: The authors present empirical results that show the superior performance of phi-1 compared to other models on coding benchmarks. They also demonstrate the effect of fine-tuning on a small dataset and highlight the emergent properties of the model.

**Weaknesses:**

1) The authors are motivated to include high-quality data (i.e., textbook). However, they also use the data generated by gpt-3.5, which may have a lot noises.  How to justify the motivation because it seems contradictory to have both high-quality and low-quality data.

2) In the caption of Figure 1, it is not light / dark green. maybe light / dark blue?

3) In section 2, the authors are motivated by: We conjecture that language models would benefit from a training set that has the same qualities as a good “textbook”: it should be clear, self-contained, instructive, and balanced.
It may be too intuitive without any justification. What about other high-quality data in addition to textbook? For example, legal documents, government document?


4) In Section 2.1, the authors rely on the GPT-4 as an annotators to filter the data. Why can we rely on it? I am aware of some studies that using GPT-4 as an annotator. However, the ability of GPT-4 on this specific task is unknown.  Justifications are needed. For example, you may want to show the comparison between the data that GPT-4 thinks high-quality v.s. low-quality.

**Questions:**

See Weaknesses.

---

> ### Author Response · Authors · 2023-11-22
> **Author response**
>
> We thank the reviewer for their time in reviewing our work and highlighting the strengths of our work on methodology and empirical findings. Please find the answers to your clarifications below.
>
> **Q. The authors are motivated to include high-quality data (i.e., textbook). However, they also use the data generated by gpt-3.5, which may have a lot noises. How to justify the motivation because it seems contradictory to have both high-quality and low-quality data.**
>
> While GPT-3.5 is far from perfect and has noises, the comparison point here are random code/text snippets from web which even after filtering, which typically have much higher levels of non-educational content. We find that with the right prompting, especially for simple coding problems the quality of GPT-3.5 generated data has far more educational value compared to a typical random text from Github repos (which form the large part of web data) or StackOverflow. The harder issue to tackle with using Synthetic data is that the generated text might not have sufficient coverage of rarer and more complex use-cases. This is why we use the mix of web-filtered data with synthetic data targeted to address specific shortcomings we observe in the webdata. We also note that our prompts were tailored to the generating model (GPT3.5)’s known strengths and weaknesses. Taking the example of synthetic textbooks, these were designed to cover the entirety of beginner programming topics, but at the same time, the topics selected were fine-grained enough that GPT-3.5 can reliably generate high accuracy text. We ensured diversity through variations in format and target audience. Of course, if we use a more powerful GPT-4 model for our generations the quality of models will very likely improve further, but generating data at our scale using GPT-4 is significantly more expensive and slower (due to access limits).
>
> **Q. In the caption of Figure 1, it is not light / dark green. maybe light / dark blue?**
> Thank you for pointing this out, this was a typo (we forgot to edit text after changing the figure) and we will fix it.
>
> **Q. In section 2, the authors are motivated by: We conjecture that language models would benefit from a training set that has the same qualities as a good “textbook”: it should be clear, self-contained, instructive, and balanced. It may be too intuitive without any justification. What about other high-quality data in addition to textbook? For example, legal documents, government document?**
>
> We certainly considered official sources of coding textbook data, there are two issues we encountered: (a) much of such high-quality material including real textbooks are copyrighted material, (b) the size of the datasets we managed to obtain while looking for such sources was very small (<<100MB) that it did not end up making much of a difference.
>
> **Q. In Section 2.1, the authors rely on the GPT-4 as an annotators to filter the data. Why can we rely on it? I am aware of some studies that using GPT-4 as an annotator. However, the ability of GPT-4 on this specific task is unknown. Justifications are needed. For example, you may want to show the comparison between the data that GPT-4 thinks high-quality v.s. low-quality.**
>
> Indeed this is an imprecise measure. We rely on it only after many iterations of manual quality checks and prompt refinements – we had small batches of data scored with GPT-4 and human labelers and look at correlations and iterate on the prompts based on feedback. We provide a typical example of high and low educational value samples from The Stack in Page 4.

---

### Official Review · Reviewer_QdXo · 2023-11-02

**Soundness:** 2 fair
**Presentation:** 3 good
**Contribution:** 2 fair
**Rating:** 5
**Confidence:** 4

**Summary:**

This paper demonstrates that by meticulously curating high-quality data, one can use significantly smaller training corpora and fewer parameters to train a model that surpasses larger models and various Python code benchmarks. Specifically, the authors employ model-based classifiers distilled from LLMs to create a compact subset of common code corpora that are more diverse, less repetitive, and generally more informative and self-contained. Moreover, they utilize another LLM to generate high-quality pretraining data and a limited set of finetuning data. The performance of the resulting 1.3B model on common benchmarks supports the claim that enhancing models may not always require more compute and data, but can be achieved using smaller amounts of high-quality data.

**Strengths:**

This paper adds to the expanding body of work that underscores the benefits of using quality, and broadens the empirical evidence to the field of LMs for code. The results presented in the paper are noteworthy; with a comparatively small model, they manage to surpass much larger models that were trained on more tokens. Another contribution of the paper is the utilization of existing LLMs to produce quality synthetic data at scale for pretraining the smaller model.

**Weaknesses:**

The paper posits the claim that meticulously curated data, combined with generating synthetic training data, can train smaller models that surpass larger ones. However, there are significant gaps regarding the creation of the training data. Specifically, the authors deliberately omit certain details, pointing to other papers that have taken a similar approach (as mentioned in footnote 1 on page 2). While the decision to withhold such details lies within a researcher's discretion, assessing the paper as a standalone piece makes it challenging to be certain that the authors' methodology doesn't introduce any data leaks potentially favoring their results. Although they explore leakage in the finetuning data, the same scrutiny isn't applied to their synthetic data. The approach taken to assess leakage against the finetuning data is somewhat nebulous. In their n-gram overlap analysis, they offer no rationale for their choice of 13-grams to gauge overlap, quickly concluding that “n-grams are not refined enough to find similar code snippets”. Subsequently, they switch to embeddings, commenting, “We observe that the embedding distance effectively captures code pairs,” without substantiating this claim.

Another point of contention is the authors' assertion that phi-1 consumed less compute for training. They overlook the computational resources expended in creating their training data, and more importantly, the compute required to train the foundational LLMs. While this doesn't directly undermine the premise that a lesser-compute model can theoretically surpass larger counterparts, it does temper its implications given that generating such data on a grand scale necessitates the distillation of more resource-intensive models.



Presentation comments for the authors (not a weakness, but should be addressed prior to final publication):
1. In the paper, all references are given using \citet, while the expected style calls for using \citep whenever the authors or the publication are not included in the sentence. See https://iclr.cc/Conferences/2023/CallForPapers for style information.
2. In page 2, you mention that you confirm the hypothesis that the number of parameters plays the key role in emergence. There is much work that show the role of parameter count in performance (Kaplan et al., 2020; Hernandez et al., 2021; Tay et al., 2021, 2022, Hoffmann et al 2022). However, there is also notable evidence against the emergence hypothesis (see Schaeffer et al 2023 and subsequent works). Since this work only contains two model sizes, I believe the language “confirm the hypothesis” is too bold, as there is no way to differentiate a sudden jump from a smooth transition over a simple line.
3. In page 6 §3.2, you say “we demonstrate that finetunuing on CodeExercises unexpectedly improves the model’s ability to use external libraries…”. Once again, I think this phrasing is not supported by the evidence given in the paper. Not only is the evaluation qualitative only, there is a single example in the paper to demonstrate that. Moreover, from the example it is clear that the model did see PyGame in the pretraining data and even phi-1-base knows how to use it. Instruction tuning during the finetuning is meant to help the model generalize to using its stored knowledge better, and there is nothing unexpected about the fact that the instruction following is better after it.
4. In the bottom of page 7, you compare phi-1 to StarCoder on the new evaluation dataset and say “phi-1 again achieves a score significantly higher than StarCoder”. If I understand table 2 right, the difference between them is 51% vs 52% and there are no statistical measures to judge the significance of the difference, and thus I believe that the phrasing “significantly higher” is misleading.
5. The paper repeatedly suggests that phi-1 outperforms larger models despite its smaller size. However, as noted by you in the conclusions, phi-1 is trained only on python code while models such as StarCoder are considerably more general, and thus the comparison is not direct. I believe this point should be made more clearly in the introduction to avoid misleading claims.
6. In the begging of “More related works” paragraph, I think you used “recent program” by mistake instead of e.g. “recent trend”.

**Questions:**

1. I would like to receive more information on how the prompts to generate the synthetic data were created at scale.
2. What verification, if any, was performed to rule out that the synthetic pre-training data generated is not highly similar to the existing samples in HumanEval?
3. In Table 2, it looks like for all prior models as well as phi-1-base (the model without finetuning), there is a significant gap between the new score and the HumanEval one. However, in both finetunened phi-1 models this gaps is removed. Is it not possible that this means that while the finetuning data may be unrelated to the new evaluation, it contains considerable leakage with HumanEval? And if so, comparing the scores of StarCoder and phi-1 on the new evaluation may be more informative, thus concluding they perform similarly?

**Details Of Ethics Concerns:**

No ethics concerns in the paper.

---

> ### Author Response · Authors · 2023-11-22
> **Author response 1/2**
>
> We thank the reviewer for highlighting the strengths of our work on demonstrating the benefits of quality and the performance of the resulting model.
>
> **Q. However, there are significant gaps regarding the creation of the training data. … assessing the paper as a standalone piece makes it challenging to be certain that the authors' methodology doesn't introduce any data leaks potentially favoring their results.**
>
> We acknowledge this as a limitation of our work for academic publication. We have added as much detail in the paper as we thought were appropriate to balance between proprietary interests and openness to research community. We additionally release our model artifact for further research, exploration, and independent evaluation. While this is not ideal or 100% transparent, we nevertheless believe that our paper captures important aspects of LLM training and is important for the research community. As noted in the paper it is also not unprecedented in our research community where highly influential papers have not disclosed dataset details (e.g. Minerva which also attribute their main contribution to be data).
>
> **Q. The approach taken to assess leakage against the finetuning data is somewhat nebulous. In their n-gram overlap analysis, they offer no rationale for their choice of 13-grams to gauge overlap, quickly concluding that “n-grams are not refined enough to find similar code snippets”. Subsequently, they switch to embeddings, commenting, “We observe that the embedding distance effectively captures code pairs,” without substantiating this claim.**
> and
> **What verification, if any, was performed to rule out that the synthetic pre-training data generated is not highly similar to the existing samples in HumanEval?**
>
> Many standard datasets in literature perform decontamination by using exact substring matches or N-grams matches to remove test set from training set: exact substring (extreme version of N-gram with N=len(human-eval-question)) was used in The Stack dataset which is the defacto standard dataset for code models; one of the largest NLP datasets DOLMA (https://blog.allenai.org/dolma-3-trillion-tokens-open-llm-corpus-9a0ff4b8da64) uses 13-gram like ours. In our 13-gram detection we did not detect any true-positive contamination in over 100k code snippets of CodeExercises (we had 4 false positives and all else were negative). Given the widespread use of N-grams for decontamination, we do not consider our conclusion unfounded.
>
> That said since we were ourselves not satisfied with this common procedure which is why we go beyond this standard practice and use other techniques for determining code-similarity, we show how code-snippets that are similar in principle are not detected by N-gram but are captured by our metrics.
>
>
> **Q. Although they explore leakage in the finetuning data, the same scrutiny isn't applied to their synthetic data.**
>
> We were able to do a thorough decontamination analysis for HumanEval on our CodeExercises dataset as both datasets were of the same type (short python functions) and are self-contained code snippets – this meant that we can devise robust versions of similarities beyond string matching. We note that many decontamination procedures in the literature (including The Stack dataset) resort to string-matching or min-hash matching against a test set which we show is not very robust (e.g., our analysis picked up similarities that N-gram analysis did not).
>
> On the other hand, our synthetic textbook is a fundamentally different NLP-heavy dataset. The code snippets in the text are often very short and used mostly for illustrative purposes: e.g, exercises or usage specific functions or structures like print and if-then, class definition etc. These are not directly comparable to HumanEval questions which have specific function level algorithmic style. Thus, we did not find this dataset suitable for in-depth ast-style analysis.

---

> > ### Author Response · Authors · 2023-11-22
> > **Author response 2/2**
> >
> > **Q. Another point of contention is the authors' assertion that phi-1 consumed less compute for training. They overlook the computational resources expended in creating their training data, and more importantly, the compute required to train the foundational LLMs.**
> >
> > This is a very fair point that we did not think about and thank you for pointing it out. We will clarify the hidden costs of our model required for creating “good quality” training data. Thank you also for noting that this alone does not undermine the results in our work. We wanted to couple additional points on this note: (a) it would be fairer to consider the amortized cost of building the foundation models (GPT4, GPT3.5) but these estimates are hard to get before their widespread use; (b) even without GPT-3.5 generation (no Synthetic textbooks, and no finetuning), our 350M parameter model trained on filtered data alone achieves ~18% performance on HumanEval (in comparison, same model trained with unfiltered data for 3x longer only achieves 12%) – we did not run this combination on 1.3B scale that our other base model is at. In this case, the only additional overhead our classifier and small number of GPT-4 annotations for quality.
> >
> > **Q. Comments on presentation.**
> > Thank you for the many suggestions on improving the presentation. We will fix all of them.
> > - Citation style: We will fix the citation style
> > - Discussion of emergence: We agree that with two model sizes our statement is strong, and will tone down to say emergence could be one possibily on why we see surprising behavior at 1.3B that were not noticeable in 350M.
> > - **Claims about StarCoder**: We did not intend to diminish the importance of more broad scoped code models like StarCoder in any way and mention this in the Limitations section multiple times. We apologize for not making this point clearer in the main paper and we will reword and clarify that our statements about performance comparision are wrt narrow scope that we study in the work. *We also agree with the reviewer that based on Table 2, it is more appropriate to conclude that StarCoder performance as comparable phi-1 (albeit in a smaller model size) and will note so in the paper.*
> > - **Improvement from finetuning**: Here we disagree with the reviewer partially – we agree that our exploration of improved API use is qualitative and will remove any quantitative/rigorous statements about it. But we disagree with the reviewer that the finding that our specific finetuning improves API usage performance is not surprising. Our finetuning is not what would be commonly considered “instruction tuning”. If anything since CodeExercises only contained algorithmic codes that DO NOT use complex APIs, our prior expectation was that the base model performance would degrade after finetuning and we found it surprising that it indeed improved significantly. Also while we provide only one example in the main paper, we show additional examples in the appendix.

---

### Official Review · Reviewer_71cs · 2023-11-05

**Soundness:** 3 good
**Presentation:** 3 good
**Contribution:** 4 excellent
**Rating:** 8
**Confidence:** 5

**Summary:**

This work presents a data and parameter efficient approach to training language models that results in small (1.3B) parameter models that are competitive and/or outperform models that are an order of magnitude larger. They propose the creation and curation of (relatively) smaller but high-quality (i.e. text-book quality) datasets for the training of high performance models, in this case, they create a 7B token dataset. The authors conjencture that LMs should learn from the same quality datasets that a human would use, and in particular that suchh datasets should be clear, self-contained, instructive and balanced.

**Strengths:**

This paper proposes a relatively high efficiency approach to training LLMs. It is impressive to get such results from a model trained on 8 A100s in 4 days. This makes this work approachable from academic research labs, which takes a step towards reversing the trend of pursuing ever larger datasets with larger models and computational requirements. This also has implications for energy efficiency and sustainability.

The paper itself is well-written and quite clear.

The authors have commited to the release of the "model for usage and evaluation by the broader community". This is important for open science and a big strength.

**Weaknesses:**

Key details on the data generation process are not shared for "proprietory reasons", yet this is central to the paper, its proposition and its results.

The lack of a broader impacts section weakens this paper.

If this weakness is addressed, I am happy to improve my score.

**Questions:**

Would you argue that the structure and compositionality of programming code is critical for achieving such high levels of performance from small models and datasets. I wonder if a takeway is to always seed training from well structured and commented code? If so, would this translate to other programming/instruction like language tasks? e.g. recipes, etc..

Do you think that the code excercises factor into improving performance by encouraging the model to self-critic whatever it generates, given that it seems to append excercises to generations?

Where do you think this approach is most likely to fail? With what kinds of tasks/data?

Does phi-1 have ICL abilities? And are there other emergent capabilities that can be forseen and easily tested?


Update: Authors have addressed my questions.

---

> ### Author Response · Authors · 2023-11-22
> **Author response**
>
> We thank the reviewer for their time reviewing the paper and highlighting our strengths in efficiency, advantages to the research community, and clarity. Below we address the questions raised in the review.
>
> **Q. Key details on the data generation process are not shared for "proprietory reasons", yet this is central to the paper, its proposition and its results.**
>
> We acknowledge this limitation of our work for academic publication. We have added as much detail in the paper as we thought were appropriate to balance between proprietary interests and openness to research community. We additionally release our model artifact for further research, exploration, and independent evaluation. While this is not ideal, we nevertheless believe that our paper captures important aspects of LLM training and is important for the research community. As noted in the paper it is also not unprecedented in our research community where highly influential papers have not disclosed dataset details (e.g. Minerva which also attribute their main contribution to be data, and GPT-3 are notable examples).
>
> **Q. The lack of a broader impacts section weakens this paper.**
>
> We have discussed some of the practical implications of the work in the Conclusions section. We will additionally discuss the environmental benefits of training small models and the opportunities for mechanistic understanding experiments. We will also clarify (based on Reviewer QdXo’s comments) that some of the hidden cost include the amortized cost of the building GPT3.5, GPT4 in the first place that can be used in turn for targeted generation. If there are other issues/benefits that the reviewer would like us to address, we are happy to add them to the paper.
>
> **Q. Would you argue that the structure and compositionality of programming code is critical for achieving such high levels of performance from small models and datasets....**
>
> The composability and structure of code generation was certainly helpful for us in effectively selecting high quality data through filtering and generation. For example, it is intuitively easier to define specifications for filtering existing web-datasets for good quality algorithmic code compared to more imprecise and subjective skills such as common-sense reasoning. We can build our textbooks with sufficient coverage based on what we consider beginner level syllabus for programming and ensure coverage.
>
> There is nothing special about python itself, so we don’t have a reason to believe why the same methodology will not extend to other languages and structured tasks. Even for unstructured and subjective tasks like common-sense reasoning, while it might be harder to filter and generate for what would constitute a “textbook” quality data, it is not impossible and can be beneficial to think of creative ways to increase the signal for reasoning skills in the training corpus.
>
> **Q. Do you think that the code exercises factor into improving performance by encouraging the model to self-critic whatever it generates, given that it seems to append exercises to generations?**
>
> This is a possible hypothesis we have not tested yet but can be studied as part of mechanistic understanding of how models learn to code using our model as a base model.
>
> **Q. Where do you think this approach is most likely to fail? With what kinds of tasks/data?**
>
>  We discuss the limitations of our model in the appendix and in the conclusions. One common pattern of limitations we see is that the model as-is does poorly on tasks that are easier with some form of memorization. For example, in terms of api usages, since our model is not trained on all of python code available on the web, it has little to no knowledge of less known libraries. We believe this can be remedied in the future by appropriate continual training where we can add additional knowledge incrementally.  Additionally, as noted earlier, there are some skills (e.g., common sense reasoning) where filtering or generating "textbook" quality data is more subjective and will require creative new ideas.
>
> **Q. Does phi-1 have ICL abilities? And are there other emergent capabilities that can be for seen and easily tested?**
>
> Since we don’t have concrete benchmarks for all types of coding abilities, our explorations in this space are more qualitative – we share some of the findings on emergent abilities in Section 3 and in Appendix.

---

### Official Review · Reviewer_zVvy · 2023-11-06

**Soundness:** 3 good
**Presentation:** 3 good
**Contribution:** 3 good
**Rating:** 5
**Confidence:** 3

**Summary:**

The paper introduces a phi model for code generation. With 1.5B parameters trained for 4 days on 8 A100 GPUs, phi-1 achieves high accuracy on code generation benchmarks like HumanEval and MBPP. The key to phi-1's performance is its training data. Instead of standard code data, it is trained on "textbook quality" data including synthetic textbooks generated by GPT-3.5 and filtered code from the web. This high quality data allows phi-1 to match larger models trained on much more data.

The authors claim that despite its small size, phi-1 displays emergent capabilities like managing complex algorithms and using external libraries not in its training data.  The authors also show that aggressive pruning of training data similar to the HumanEval test set shows phi-1's performance boost is not due to "contamination". phi-1 outperforms StarCoder even after removing 40% of training data. The authors argue high quality data is key to advancing language models. Creating diverse, high quality datasets remains challenging and important for future work.

**Strengths:**

+ The paper achieves state-of-the-art results on code generation benchmarks with a much smaller model trained on far less data. This re-emphasizes the power of high quality, tailored training data.
+ The model requires less compute resources for training compared to larger models.
+ The paper also shows rigorous evaluation of potential training set contamination.

**Weaknesses:**

- The paper has low novelty, and the importance of data quality in LLMs is well known.
- The paper is only evaluated mainly on short Python functions. How would it perform on more complex, real-world coding tasks.

**Questions:**

- How was the quality of the synthetic textbook data evaluated? What measures were used to ensure it is high quality and diverse?
- How do the results of synthetic data, and using textbooks generalize to other domains beyond code? Does this approach advance the scientific understanding of training language models for code generation?
- What are the unique contributions compared to prior work on tailored training data and prompted training?

---

> ### Author Response · Authors · 2023-11-22
> **Author response**
>
> We thank the reviewer for their time and effort in reviewing our paper and highlighting the strengths of computational benefits and decontamination analysis. Now we address the main concerns raised in the review.
>
> **Q. The paper has low novelty, and the importance of data quality in LLMs is well known. &
> Q. What are the unique contributions compared to prior work on tailored training data and prompted training?**
>
> On this point, we respectfully disagree with the reviewer about the lack of novelty in our paper. While the philosophy of good quality training data being good for learning is not new, the predominant trend in LLMs has been on scaling-up data and model sizes, as evidenced by numerous papers on scaling laws. To our knowledge, our paper is among the first to provide strong and compelling evidence on the effectiveness of high-quality data to achieve state-of-the-art performance in LLMs. The novelty in our work also lies in specific approaches we took to generate and use "textbook quality" data. Our paper contributes uniquely in several ways:
> - Targeted synthetic data: We show the effective usage of targeted synthetic data to address specific gaps in existing data sources: e.g. our synthetic textbooks provide useful signals for the model by enhancing signal from natural language text that explains simple algorithms in detail.
> - Filtering methodology: We used a combination of GPT-4 (frontier LLM), a small LM classifier, and human-in-the-loop for textbook-quality data filtering. Such methodologies incorporating frontier LLMs strategically in the loop are in the early stages of experimentation and our paper contributes to it. We believe identifying best practices in this area is significant for future models.
> - Overall, our model achieves top-tier performance on algorithmic code generation (next only to latest GPT models) despite being 10x-100x smaller in size compared to the other top models. This provides strong evidence that selecting high quality data in creative ways is key to unlocking efficient learning.
>
> **Q. The paper is only evaluated mainly on short Python functions. How would it perform on more complex, real-world coding tasks.**
>
> We intentionally limited our model's scope (at the beginning of the project) to Python programming on algorithmic tasks. This choice was guided by our hypothesis that high-quality data is more critical to learn reasoning and planning-based skills, which are arguably among the hardest to learn from data scaling alone. Nevertheless, we discuss qualitative examples in the paper show that despite our narrow focus, our model was able to perform well on broader tasks of using popular APIs and understanding general natural language formats. We also discuss the limitations of the models in the appendix. Note that we use qualitative evaluations in these sections as there are no standard benchmarks covering broad use-cases for code generation.
>
> **Q. How was the quality of the synthetic textbook data evaluated? What measures were used to ensure it is high quality and diverse?**
>
> For our synthetic datasets, we primarily relied on manual evaluation to assess quality and diversity. We used the performance of the resulting (small) model when trained on representative examples of generations as quantitative validation for our manual quality checks. We note that the evaluation of quality and diversity is subjective and depends on the specific task at hand; as such, there are no good metrics to gauge utility for LLMs. We considered these difficulties and the limitations of manual evaluation when crafting prompts for the generation process. The prompts were tailored to the generating-model's (GPT3.5) known strengths and weaknesses.  Taking the example of synthetic textbooks, these were designed to cover the entirety of beginner programming topics, but at the same time, the topics selected were fine-grained enough that GPT-3.5 can reliably generate high accuracy text. We ensured diversity through variations in format and target audience.
>
>  **Q. How do the results of synthetic data, and using textbooks generalize to other domains beyond code? Does this approach advance the scientific understanding of training language models for code generation?**
>
> Our paper focuses on code generation as a proof of concept to demonstrate the benefits of using textbook-quality data in training LLMs. This serves as a compelling standalone use-case and a representative example for other tasks that require reasoning-based skills. We are aware of similar follow up attempts in more general language understanding and reasoning tasks.

---

### Meta-Review · Area_Chair_kRaZ · 2023-12-09

**Metareview:**

In this paper, the authors introduced phi-1, a new code LM which, despite the small number of model parameters, can outperform many strong and larger model baselines. The main contribution of the work is the collection of “textbook quality” public data and synthetically generated textbooks and exercise data (generated by LLMs).

The reviewers appreciated the contribution of the textbook data collection and the impressive model performance on popular benchmarks like HumanEval and MBPP (despite the smaller model size and small compute cost). However, some major concerns remain, including: (i) the concerns about data quality (including the potential data leakage) in both public data and synthetic data; and (2) limited experimental results on 2 benchmarks for short/basic Python code generation tasks only. Since the paper's main contribution is about the textbook quality dataset, I highly encourage the authors to make a more significant part of the paper about the data quality (e.g. more transparent data collection/generation process, data distribution and analysis including human evaluation, and more detailed comparisons between training and test data).

**Justification For Why Not Higher Score:**

The paper's main contribution is a curated dataset of textbook-quality samples. However, the authors did not fully describe and explain many important aspects of the dataset, including the potential data noise/leakage to the test datasets. It is therefore quite hard to judge the contribution of the paper and whether the proposed dataset could be replicated and generalized to other models.

**Justification For Why Not Lower Score:**

N/A

---

### Decision · Program_Chairs · 2024-01-16

Reject